# Global and Regional Myocardial Work in Female Adolescents with Weight Disorders

**DOI:** 10.3390/jcm10204671

**Published:** 2021-10-12

**Authors:** Justine Paysal, Etienne Merlin, Emmanuelle Rochette, Daniel Terral, Stéphane Nottin

**Affiliations:** 1LAPEC EA4278 Laboratory, Avignon University, F-84000 Avignon, France; jpaysal@chu-clermontferrand.fr; 2Néonatologie et Réanimation Pédiatrique, CHU Clermont-Ferrand, F-63000 Clermont-Ferrand, France; 3Pédiatrie, CHU Clermont-Ferrand, F-63000 Clermont-Ferrand, France; e_merlin@chu-clermontferrand.fr (E.M.); e_rochette@chu-clermontferrand.fr (E.R.); dterral@chu-clermontferrand.fr (D.T.); 4INSERM, CIC 1405, Unité CRECHE, Université Clermont Auvergne, F-63000 Clermont-Ferrand, France

**Keywords:** global myocardial work, regional myocardial work, loading conditions, left ventricular strains, weight disorders, female adolescents

## Abstract

Background: Anorexia nervosa (AN) and obesity (OB) lead to changes in SBP (i.e., loading conditions) that may affect left ventricular (LV) myocardial work (MW). The novel concept of LV pressure-strain loops allows non-invasive estimation of MW, this latter being correlated with cardiac energy metabolism. In addition, the study of regional MW can detect subtle alterations in cardiac function by highlighting an abnormal distribution of MW. Objective: The aim of this study was to assess the cardiac function of AN and OB patients by evaluating global and regional LV strains and MW. Methods: Eighty-seven female adolescents, comprising 26 with AN (14.6 ± 1.9 yrs. old), 28 with OB (13.2 ± 1.4 yrs. old), and 33 controls (14.0 ± 2.0 yrs. old) underwent speckle-tracking echography to assess global and regional LV strains and MW. Results: SBP was higher in adolescents with obesity than in AN patients or controls. Global MW was similar between groups. In AN patients and controls, longitudinal strains were higher at the apex than at the base of the LV, whereas they were similar in obesity patients, owing to a decrease in their apical longitudinal strain. Consequently, their MW was higher at the basal level than either of the other two groups (1854 ± 272 vs. 1501 ± 280 vs. 1575 ± 295 mmHg% in OB patients, AN patients, and controls, respectively. Conclusion: Despite altered SBP, the global MW of adolescents with weight disorders was unaffected. However, in adolescents with obesity, the distribution of their regional LV MW was altered, which might reflect specific regional remodeling.

## 1. Introduction

Adolescent weight disorders are extremely common [1,2], ranging from anorexia nervosa (AN), defined by a significant weight loss [1,3] to obesity (OB), characterized by excess weight [2,4]. Their impact on the cardiovascular system is well-established. In AN, cardiac complications are frequent, reaching 80% in some studies. They range from morphological cardiac abnormalities to electrical abnormalities with a potential risk of sudden death [5]. In large epidemiological studies, obesity is associated with increased incidence of heart failure. Analysis of the Framingham Heart Study revealed that obese individuals had twice the risk of heart failure over a mean follow-up of 14 years [6]. Whereas their left ventricular (LV) systolic and diastolic functions seemed preserved [1,2,7], the opposite effects have been observed on arterial blood pressure (BP), with lower systolic BP (SBP) in AN patients [7] and conversely higher values in those with obesity [8]. Alteration of SBP and thus afterload in these pediatric populations probably induces compensatory adjustments of their myocardial performance to maintain a normal ejection fraction (EF).

Myocardial strain analysis from speckle-tracking echocardiography (STE) has emerged in the last decade as a reliable tool for studying myocardial mechanics, adding information on cardiac performance compared with traditional variables of LV function such as EF [9]. However, these indices have been limited by their inability to account for changes in afterload [10]. Recently, the non-invasive estimation of global myocardial work (GMW) by the assessment of LV pressure-strain loops (LV-PSL) has overcome this limitation by taking into account the loading conditions [11]. GMW, which reflects myocardial performance [12,13], cardiac glucose metabolism [14] and oxygen consumption [12], potentially enhances myocardial function assessment.

Another important feature of STE analysis is that it offers a regional assessment of LV strains and MW. In the normal heart, it is well-established that LV strains are higher at the apex than in middle and basal regions [15,16]. In adolescents with obesity, very few studies have assessed regional myocardial strains. Mangner et al. [17] reported lower LV strains at the basal-septal level from tissue Doppler analysis, whereas Binnetoglu et al. [18] observed lower LV strains only at mid- and apical segments. In AN patients, Morris et al. [19] reported a decrease in LV apical strains, but only in five patients with purge behavior. Taken together, these findings precluded any firm conclusion regarding regional strain alteration in adolescents with weight disorders. Moreover, regional assessment of MW is lacking and remains to be addressed in these adolescents. It could provide additional information to detect early subtle myocardial remodeling [11,20].

This study set out to describe global and regional LV strains and MW in adolescents with weight disorders. We hypothesized that: (I) GMW would be reduced in AN patients secondarily to a decrease in SBP and conversely increased in OB patients owing to an increase in SBP; and (II) differences in the distribution of regional MW would be present in adolescents with weight disorders, compared with controls, as a result of regional alteration of strains and adverse LV remodeling.

## 2. Methods

### 2.1. Study Population

This prospective study included female adolescents with anorexia nervosa (AN patients, *n =* 30), normal weight (*n =* 34), and obesity (OB patients, *n =* 30) aged 10–18 years. The patients had been diagnosed in a pediatric department of a university hospital in France between March 2019 and January 2020. The duration of the disease was evaluated from the study of their body mass index (BMI) curves. AN patients fulfilled the DSM V criteria for AN (American Psychiatric Association) [21] and those with obesity met the IOTF C30 criteria [22]. All the OB patients were recruited in a specific establishment for obese adolescents, at the beginning of their 1-year medical management and ongoing monitoring by medical staff. AN patients were recruited at the hospital, either at the beginning of their medical care (*n =* 16) or during the first months of the refeeding period (*n =* 10). None of the AN adolescents reported purge and/or binge behavior or had refeeding syndrome, as evidenced by their normal ionogram. The adolescents with obesity had primary obesity (i.e., non-syndromic or not secondary to endocrine disorder) and did not present diabetes or dyslipidemia (verified on the same day as echocardiography by a blood test). They were not diagnosed with arterial hypertension. The BMI *z* score was calculated for all participants. None had chronic disease, congenital heart defects, or positive family history of cardiac disease. Written informed consent was obtained from the study participants and their guardians. The Ile-de-France Ethics Committee approved the protocol for this study (18.12.05.66738 CAT 2).

### 2.2. Anthropometric and Clinical Assessments

Body height and body mass were measured. BMI was calculated as body mass.body height^−2^. BPand resting HR were measured using an automatic device (General Electric, Dynamap PRO 300 V2, Boston, MA, USA). A fasting venous blood sample was taken for biochemical assays, with in particular NT-proBNP by automated immunoassay. 

### 2.3. Echocardiographic Recordings

Echocardiography was carried out with the subject in the left lateral decubitus position, with Vivid ultrasound systems (GE Healthcare, Horten, Norway) using a 3.5 MHz transducer (M4S probe). Cine loops were recorded in parasternal long axis and apical (5, 4, 3 and 2 chamber) views and saved for blinded offline analysis (EchoPac, BT203 version, GE Healthcare). Grayscale images were saved at a frame rate of 80–90 frames/sec and color tissue velocity images at a frame rate of 120–140 frames/s. 2D echocardiographic measurements were made in accordance with the American Society of Echocardiography guidelines [23]. All echocardiographic data were averaged from measurements obtained on 3–5 cardiac cycles.

### 2.4. Cardiac Morphology

LV diameters and myocardial thickness were measured from the parasternal long axis view. LV mass was estimated using the Devereux formula and indexed to height^2.7^ as recommended in the pediatric population [24,25]. Relative wall thickness (RWT) was calculated by the formula: (2 × posterior wall thickness)/LV end diastolic diameter. LV volumes were assessed using Simpson’s biplane method.

### 2.5. Left Ventricular (LV) Systolic and Diastolic Functions

LV diastolic function was assessed from peak early (E wave) and atrial (A wave) transmitral flow velocities. EF was assessed using Simpson’s biplane method. Global longitudinal strain (GLS) was obtained as previously detailed [26] and a regional analysis was carried out to obtain basal, median, and apical LS.

### 2.6. Myocardial Work Quantification

MW and related variables were estimated using the Automatic imaging function of the EchoPac software [27]. MW was estimated as a function of time throughout the cardiac cycle by the combination of LV strain data (recorded on the apical 4, 3 and 2 chamber) obtained by STE and a non-invasively estimated LV pressure curve as described and validated by Russell et al. [11,28]. Peak arterial pressure measured with a cuff manometer was assumed to be equal to peak systolic and diastolic LV pressures and to be uniform throughout the ventricle. MW was then quantified by calculating the rate of segmental shortening by differentiating the strain curve and multiplying the resulting value by the instantaneous LV pressure. A segmental analysis was performed to obtain the basal, median and apical MW. Myocardial work efficiency (MWE) was calculated as previously described [27].

### 2.7. Statistical Analyses

Statistical analyses were performed using MedCalc® Statistical Software version 20.013 (MedCalc Software Ltd, Ostend, Belgium; https://www.medcalc.org; 7 October 2021). All values were expressed as mean ± SD. One-way analysis of variance (ANOVA) was used to compare groups after checking the normality of distribution of each variable by a Shapiro–Wilk test. In the absence of normal distribution, the non-parametric Kruskal–Wallis test was used. Statistical significance for all analyses was assumed at *p* < 0.05. Intra-observer and inter-observer variability for 2D-strain analysis had been previously assessed in our laboratory, yielding maximal coefficient of variation values less than 8% for strains [29]. In the present study, the intra-observer variability for GMW was assessed on duplicate measurements in 60 subjects. The variability was very low, with a coefficient of variation of 4.4%.

## 3. Results

### 3.1. Population Characteristics and Resting Echocardiography

Eighty-seven female adolescents, comprising 26 with AN, 33 with normal weight, and 28 with obesity were included. Table 1 shows the anthropometric characteristics and the standard echocardiographic variables of our population. As expected, BMI was different between our groups. Illness duration was also different between AN and OB patients. LV wall thicknesses, mass, volume and RWT were higher in the obesity group. EF was similar between groups. The diastolic function of AN patients was characterized by a lower A wave, whereas that of OB patients was characterized by a higher E wave. NT-proBNP was higher in AN (79 ± 60 ng.L^−1^) than in OB (34 ± 23 ng.L^−1^) and in normal weight (39 ± 21 ng.L^−1^) patients (*p* < 0.01).

### 3.2. Global Longitudinal Strain and Myocardial Work

SBP, GLS and GMW are presented in Figure 1 and Table 2. As expected, SBP was decreased in AN patients and increased in OB patients compared with controls. DBP was similar between groups. We note that five adolescents with obesity had SBP, two DBP, and six both SBP and DBP above the 95th percentile for their age defining high BP. In AN patients, GLS was higher than in controls, and their LV-PSL was shifted leftward and downward, as a result of the increase in their GLS and decrease in their LV pressures. In adolescents with obesity, GLS was similar to controls and their LV-PSL was shifted upward, secondary to the increase in their LV pressures. However, the area under the loop, reflecting the GMW, was similar between groups. Similar values of GWE were also observed between groups. 

### 3.3. Regional Longitudinal Strains and Myocardial Work

Regional LS and MW are presented in Table 2 and Figure 2 and Figure 3. In AN patients, median LS was higher than in controls, while apical and basal LS were similar. There was no difference in MW whatever the region of LV concerned. We observed that their LS and MW, like that of the controls, increased from the base to the apex.

In adolescents with obesity, median and apical LS were lower, while basal LS was similar to controls. Their basal MW was higher than that of controls. Note that their LS and MW, unlike that of the controls, did not increase from the base and the apex.

## 4. Discussion

MW analysis offers the possibility of exploring LV function considering the effects of alteration in loading conditions [11]. In this study, we demonstrate that global MW was similar between AN patients, OB patients and controls, related to their concomitant alterations of BP and LV strains. Based on regional assessments, we further found that a different regional remodeling of LS and MW occurred. Whereas in controls and AN patients LS and MW increased from the base and apex, this gradient of LS and MW disappeared in OB patients, and their MW at the base of the LV was higher than in the other groups.

### 4.1. Global Myocardial Work in Anorexia Nervosa (AN) and Obesity (OB) Patients

At variance with our hypothesis, the first main result of our study was that the GMW was similar between groups. This was an unexpected finding since differences in BP were observed between our AN patients, OB patients and controls, confirming results of previous studies [7,8]. MW is a new variable that takes into account both deformation and afterload through interpretation of strain in relation to dynamic non-invasive LV pressure [12,13] (estimated using the method of Russel et al. by measurement of cuff pressures [11,13,27]). The absence of differences in GMW between our groups, therefore, results from an interplay between the estimation of intraventricular pressure and the GLS curves, as shown by LV-PSL [11,13,30].

In AN patients, SBP was reduced compared with controls, as previously observed [7]. On the other hand, they underwent a significant increase in their GLS. To the best of our knowledge, only Morris et al. (2017) have assessed LV strains, and they reported normal values in AN patients [19]. In the present study, the lower BP associated with the higher GLS altered the pattern of the LV-PSL in our AN patients. However, the area inside the loop, and thus the GMW, remained unchanged compared with controls. Since a relationship between GMW and myocardial glucose uptake has been demonstrated [10,13,27,28], AN patients may have preserved their cardiac energy metabolism despite their calorie deprivation.

Although our obesity patients were not previously known to be hypertensive, 11 of them had SBP above the 95th percentile for age, the threshold for defining arterial hypertension, and on average their SBP values were significantly higher than those of controls. Moreover, eight of them had a DBP above the 95th percentile, although average values did not reach statistical significance with controls. They had a normal GLS, a result inconsistently found since other studies described a decrease in GLS in children with obesity [17,31,32,33,34,35]. Their GMW was unchanged compared with controls. To the best of our knowledge, no previous studies have assessed MW in adolescents or adults with obesity. Of note, recent studies have assessed MW in hypertensive patients [9,30], and interestingly, Chan et al. [30] reported that GMW was increased only in grade II (i.e., with SBP > 160 mmHg), suggesting that the increase in MW is observed only for high values of BP. This could explain why in our adolescents with obesity who showed a moderate elevation of BP, GMW remained similar to that observed in controls.

### 4.2. Regional Analysis of Myocardial Work in AN and OB Patients

In healthy subjects, the LS increased from the base to the apex [15,16], probably related to regional differences in wall stress [36], LV thickness, and/or radius of curvature [15,37]. In our AN patients and controls, this gradient was also observed, with higher values observed in the apical region. As a consequence, since the same estimation of LV pressures was used for the calculation of each regional MW, the MW was also higher at the apex than at the base. However, an important finding of our study was that these LS and MW regional gradients were totally abolished in subjects with obesity. The loss of apex-to-base gradient was exclusively due to their apical SL, which was significantly reduced. However, owing to their higher BP, the apical MW remained normal compared with controls. 

Importantly, the normal LS of subjects with obesity associated with higher BP significantly increased their basal MW compared with controls. These findings underline the usefulness of assessing regional LS and MW to detect subtle alterations in myocardial function, especially in our obesity patients with normal EF, GLS and GMW. Our results in obesity patients raise the question of the significance of the specific increase in basal MW and its possible predictive value for an adverse remodeling of these regions, as suggested by previous studies in other pathological situations [20,38]. Interestingly, one study reported a specific decrease in strains of the basoseptal region in children with obesity [17]. Moreover, basal hypertrophy has been described in situations of arterial hypertension, explained by a predominant increase in wall stress and by the presence of fibrosis at the base [12,39,40]. The specific increase in basal MW could thus be predictive of LV long-term morphological and functional remodeling in obesity patients. A good reproducibility for the assessment of MW was noted, supporting the possibility of a promising application of this new tool in clinical practice [10].

### 4.3. Study Limitations 

To assess the impact of weight disorders on myocardial mechanics, we recruited AN and OB patients. We chose to analyze a female cohort because the prevalence of AN in male adolescents is very low (prevalence ranges between 0.16% and 0.3% in males, against about 3% in females). Changes in myocardial function were mainly observed in OB patients, a result potentially explained by their higher duration of illness (and also of exposure to altered loading conditions) compared with AN patients. Further studies will be needed to assess regional LV strains and myocardial work in AN adult patients with a longer duration of illness. LV intraventricular pressures, estimated from BP measured with cuff and sphygmomanometer at the brachial artery level may represent another limitation to the study of MW [11]. However, previous studies have shown that this methodology is sufficiently reliable for the context of regional MW analysis [11,13]. Finally, we assessed MW using pressure-strain loops without taking into account regional wall thickness and mid-wall curvature.

## 5. Conclusions

This study is the first to describe global and regional LV strains and MW derived from LV-PLS in adolescents with weight disorders. We found that GMW was similar in AN patients, OB patients and controls. However, our data suggest a specific redistribution of regional MW in OB, with higher values at the LV base compared with controls. Taken together, our results emphasize the importance of considering a regional assessment of LV strains and MW. This approach, taking into account not only deformations but also dynamic non-invasive LV pressure, seems a promising means to detect early subtle alterations in regional myocardial function in AN and OB patients with altered afterload.

## Figures and Tables

**Figure 1 jcm-10-04671-f001:**
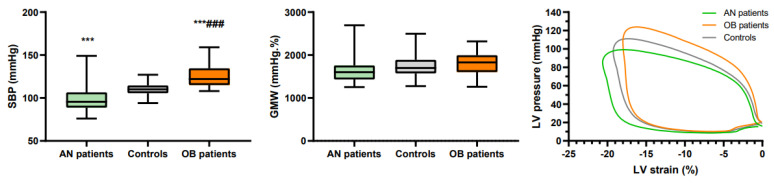
Systolic blood pressures and global myocardial work. ***: significantly different from AN patients (*p* < 0.001). ###: significantly different from controls (*p* < 0.001).

**Figure 2 jcm-10-04671-f002:**
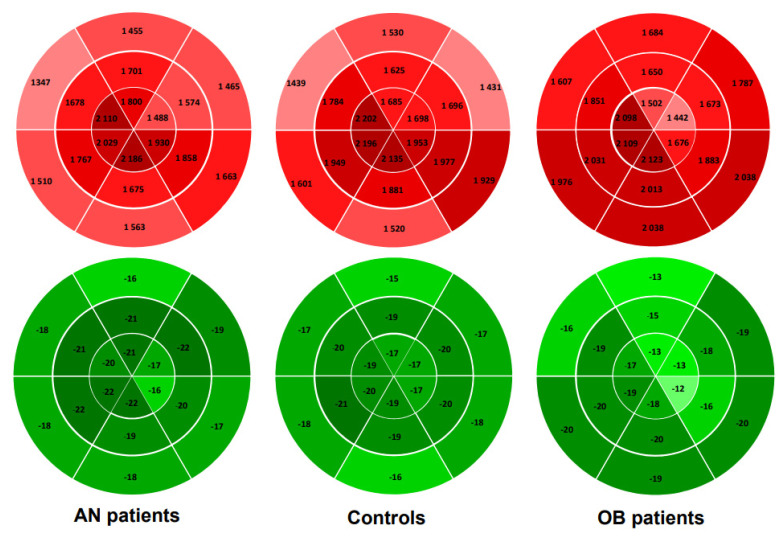
Bull’s eye plots of 18 segments of left ventricular model showing group averages in segmental myocardial work index. Apex-to-base gradient is visible in AN patients and controls, but not in obese patients.

**Figure 3 jcm-10-04671-f003:**
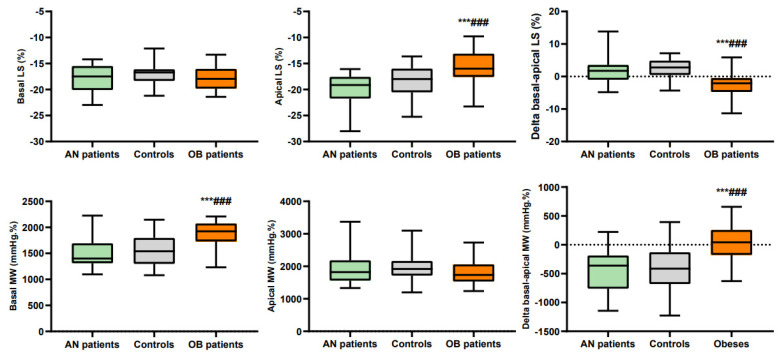
Basal and apical longitudinal strains and myocardial work. ***: significantly different from AN patients (*p* < 0.001). ###: significantly different from controls (*p* < 0.001).

**Table 1 jcm-10-04671-t001:** General characteristics and standard ultrasound variables of our population.

	AN Patients (*n =* 26)	Controls (*n =* 33)	OB Patients (*n =* 28)
Age (years)	14.6 ± 1.9	14.0 ± 2.0	13.2 ± 1.4 ^#^
Illness duration (months)	19 ± 9	-	130 ± 27 ^###^
Anthropometry			
Height (cm)	159.8 ± 9.1	162.6 ± 10.0	162.9 ± 5.7
Body mass (kg)	40.7 ± 8.2 ***	51.2 ± 9.8	90.4 ± 13.0 ***^###^
BMI (kg.m^−2^)	15.8 ± 2.1 ***	20.0 ± 3,2	34.0 ± 4.1 ***^###^
BMI *z* score	−1.8 ± 1.1 ***	0.4 ± 1.3	4.1 ± 0.6 ***^###^
Ultrasound variables			
LV septum thickness (cm)	0.72 ± 0.15	0.76 ± 0.11	0.83 ± 0.18 ^#^
LV posterior wall thickness (cm)	0.67 ± 0.12 ***	0.74 ± 0.11	0.88 ± 0.13 ***^###^
LV mass (g)	79 ± 26 ***	96 ± 24	128 ± 29 ***^###^
LV mass^2.7^ (g.m^2.7^)	22 ± 6 ***	26 ± 5	34 ± 7 ***^###^
RWT	0.33 ± 0.06	0.36 ± 0.06	0.39 ± 0.06 ***^###^
LV end-diastolic volume (mL)	79 ± 19	90 ± 23	119 ± 27 ***^###^
LV end-systolic volume (mL)	29 ± 8	33 ± 9	42 ± 11 ***^###^
Ejection fraction (%)	64 ± 5	64 ± 6	64 ± 6
E wave (cm.s^−1^)	84 ± 17	82 ± 14	90 ± 11 *
A wave (cm.s^−1^)	30 ± 6 ***	40 ± 8	39 ±7 ^###^
E/A	2.9 ± 0.9 ***	2.1 ± 0.5	2.4 ± 0.5 ***^###^

Values are mean ± SD; *: significantly different from controls (*: *p* < 0.05; *** *p* < 0.001. ^#^: significantly different from AN patients (^#^: *p* < 0.05; ^###^: *p* < 0.001). BMI: body mass index. LV: left ventricular. RWT: relative wall thickness.

**Table 2 jcm-10-04671-t002:** Blood pressure, longitudinal strains and myocardial work.

	AN Patients (*n =* 26)	Controls (*n =* 37)	OB Patients (*n =* 28)
Hemodynamic constants			
Heart rate (bpm)	56 ± 12 ***	74 ± 11	76 ± 11 ^###^
SBP (mmHg)	99 ± 14 ***	110 ± 8	124 ± 12 ***^###^
DBP (mmHg)	63 ± 11	67 ± 7	69 ± 13
MBP (mmHg)	75 ± 12 ***	81 ± 7	87 ± 12 ***^###^
Longitudinal strains			
GLS	−18.8 ± 2.0 **	−16.9 ± 2.8	−16.8 ± 1.9 ^##^
Basal LS	−17.9 ± 2.6	−16.9 ± 2.2	−17.8 ± 2.1
Median LS	−21.3 ± 2.0 ***	−20.1 ± 2.0	−18.2 ± 2.1 ***^###^
Apical LS	−20.1 ± 3.4	−18.4 ± 3.2	−15.8 ± 3.5 ***^###^
Delta basal-apical LS	2.1 ± 4.6	2.1 ± 3.7	−2.5 ± 3.6 ***^###^
Myocardial work			
GMW	1 658 ± 335	1734 ± 287	1780 ± 292
GWE	92.8 ± 3.8	91.7 ± 5.2	92.7 ± 3.7
Basal MW	1501 ± 280	1575 ± 295	1855 ± 272 ***^###^
Median MW	1709 ± 318	1819 ± 297	1850 ± 363
Apical MW	1924 ± 480	1978 ± 418	1825 ± 375
Delta basal-apical MW	−423 ± 351	−403 ± 404	30 ± 331 ***^###^

Values are mean ± SD; *: significantly different from controls (**: *p* < 0.01; *** *p* < 0.001. ^#^: significantly different from anorexics (^##^: *p* < 0.01; ^###^: *p* < 0.001). SBP: systolic blood pressure. DBP: diastolic BP. MBP: mean BP. GLS: global longitudinal strain. GMW: global myocardial work. GWE: global work efficiency.

## Data Availability

The data presented in this study are available on request from the corresponding author.

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
