# Peer review of "Global and Regional Myocardial Work in Female Adolescents with Weight Disorders"

_jcm, 2021, doi:10.3390/jcm10204671_

Round 1

Reviewer 1 Report

In this paper the authors Emphasize the possibility to assess the cardiac function of anorexia nervosa  and obesity patients by evaluating global and regional LV strains and MW.  Results: SBP was higher in adolescents with obesity than in AN patients or controls. Global MW was similar between groups. In AN patients and controls, longitudinal strains were higher at the apex than at the base of the LV, whereas they were similar in obesity patients, owing to a decrease in their apical longitudinal strain.

The Working hypotheses are well set but there are some byas :

- The calculation of the mass of the left ventricle with the Deveraux formula is not always accepted; better to rely on the MRI Assessment

  • An  non-invasively evaluation estimated LV pressure curve as described and validated by Russell et al., .......:
  • It has some flaws and it would be better to measure  an estimated LV pressure curve by an haemodinamic catheterization.

-It should be better evaluated  obesity associated with higher BP and  increased  basal MW compared with controls with normal BP

- The loss of apex-to-base gradient in obesity was exclusively due to their apical SL…. but should be studied and confirmed by MRI analysis and coronaric angiography

Author Response

In this paper the authors Emphasize the possibility to assess the cardiac function of anorexia nervosa and obesity patients by evaluating global and regional LV strains and MW.  Results: SBP was higher in adolescents with obesity than in AN patients or controls. Global MW was similar between groups. In AN patients and controls, longitudinal strains were higher at the apex than at the base of the LV, whereas they were similar in obesity patients, owing to a decrease in their apical longitudinal strain.

The Working hypotheses are well set but there are some byas :

- The calculation of the mass of the left ventricle with the Deveraux formula is not always accepted; better to rely on the MRI Assessment

In our study, we assessed LV mass of our adolescents with a non-invasive method based on echocardiographic recordings. We used the linear method, and more especially the cube formula: LV mass = 0.8 · 1.04 · [(IVS + LVID + PWT)3− LVID3] + 0.6 g, where IVS is interventricular septum; LVID is LV internal diameter, and PWT is inferolateral wall thickness. The estimation of LV mass by this method is widely used by clinicians and researchers and has many advantages, because it is simple, quick, and subject to less measurement variability compared to 2D-based measurements. There is a large body of evidence to support the accuracy of this method, and most studies that relate LV mass to prognosis are based on this method. (See for more details the recommendations of the American Society of Echocardiography and the European Association of Cardiovascular Imaging, Lang et al., 2015).

We acknowledge that MRI would have been more accurate to assess LV mass, but this method would be less appropriate in adolescent (and especially in healthy adolescents) due to logistical issues (time-consuming, second appointment), and sometimes anxiety.

An  non-invasively evaluation estimated LV pressure curve as described and validated by Russell et al., .......:

It has some flaws and it would be better to measure an estimated LV pressure curve by an haemodinamic catheterization.

You are right that the assessment of LV pressure curve directly by haemodynamic catheterization is more accurate that the non-invasive method proposed by Russell et al. (2015). However, for evident ethical reasons, cardiac catheterization is absolutely unthinkable in obeses, anorexic or healthy adolescents.

Importantly, the validity of the method proposed by Russell et al. (2015) was confirmed in the experimental part of their study by an excellent correlation and a good agreement with loop area by invasive pressure – segment length analysis. Moreover, in the clinical part of their study, LV pressure – strain loop area using the non-invasive LV pressure curve showed a strong correlation and a good agreement with loop area using invasive LV pressure.

-It should be better evaluated obesity associated with higher BP and increased basal MW compared with controls with normal BP

We are not sure to well understand your question. If your comment involves the differences of arterial pressure observed between obese and controls adolescents, this question has been addressed by the authors of this paper. However, we think that a significant increase of resting blood pressure in obese adolescent is probably one consequence of obesity and thus has not been considered as an exclusion criterion by the authors of this study in this specific population.

- The loss of apex-to-base gradient in obesity was exclusively due to their apical SL…. but should be studied and confirmed by MRI analysis and coronaric angiography

The strength of the echocardiographic evaluations is to be non-invasive, and recent advances enabled left ventricular strains and myocardial work to be evaluated. A coronary angiogram has some risks, such as radiation exposure from the X-rays used. Despite major complications are rare, potential risks and complications include injury to the catheterized artery, arrhythmia, allergic reactions to the dye or medications used during the procedure, or infection. In this context, for evident ethical reasons, coronary angiography is inappropriate in our study populations of obese, anorexic or healthy adolescents.

Reviewer 2 Report

The authors investigated global and regional myocardial work and longitudinal strain by transthoracic echocardiography in female pediatric patients with anorexia nervosa (n=26), obesity (28), and normal controls (n=33). Global myocardial work was similar between the groups. In regional analysis, longitudinal strain was lower in the apex in the obesity group when compared to controls and anorexia patients. Myocardial work was higher at the basal level when compared to the other groups.

With myocardial work, there is a promising new software analysis available to evaluate left ventricular systolic function. It is of great interest to investigate different diseases with this new tool to find clinical applications and possibly detect subtle LV dysfunction.

The authors of this manuscript chose to investigate a cohort of pediatric patients with obesity and anorexia. While their investigation is novel, there are some major issues that need to be addressed.

  • The main problem for regional strain analysis (and thereby also regional myocardial work) is image quality. The authors should address this issue, particularly considering their cohort of pediatric patients with anorexia and obesity, those two extremes where transthoracic echocardiography comes to its limits. What about image quality in the included patients? Was it possible to analyze all three apical views in all included patients? Were the results checked for intra- and inter-observer variability? Who were the examiners (one or several, level of expertise?) who recorded the images?

  • Line 128: There is a large different in duration of illness between the anorexia and obesity group (19 vs 130 months). As the authors write later, myocardial work reflects myocardial remodeling. If remodeling is one of the main targets of this analysis, the examined two diseases should have been in place for the same time, otherwise duration of illness can be suspected to be a significant confounder. Please comment on this.

Minors:

  • Lines 86f: please specify which ultrasound systems were used.
  • Line 33: the well-established „impact“ on the cardiovascular system should be further addressed here: morbidity/mortality, long term effects.
  • What was the clinical state/ condition of the patients at time of inclusion? Outpatient clinic, inpatients? Were any of the anorexia patients on artificial nutrition? Refeeding syndrome?
  • Were there any blood results to be reported, NTproBNP?
  • What types of anorexia were included, purge/binge etc.?
  • Why did the authors choose to analyze a female only cohort, please comment on this.
  • Lines 127/128 „ as expected BMI was different between our groups”. This sentence is redundant as patients with anorexia and obesity were included.
  • Abstract, line 25, „reflecting specific regional myocardial remodeling“: this is only one possible explanation. Considering the small number of included patients the message should be phrased differently, e.g. “might reflect specific regional remodeling”.

Author Response

The authors investigated global and regional myocardial work and longitudinal strain by transthoracic echocardiography in female pediatric patients with anorexia nervosa (n=26), obesity (28), and normal controls (n=33). Global myocardial work was similar between the groups. In regional analysis, longitudinal strain was lower in the apex in the obesity group when compared to controls and anorexia patients. Myocardial work was higher at the basal level when compared to the other groups.

With myocardial work, there is a promising new software analysis available to evaluate left ventricular systolic function. It is of great interest to investigate different diseases with this new tool to find clinical applications and possibly detect subtle LV dysfunction.

The authors of this manuscript chose to investigate a cohort of pediatric patients with obesity and anorexia. While their investigation is novel, there are some major issues that need to be addressed.

The main problem for regional strain analysis (and thereby also regional myocardial work) is image quality. The authors should address this issue, particularly considering their cohort of pediatric patients with anorexia and obesity, those two extremes where transthoracic echocardiography comes to its limits. What about image quality in the included patients? Was it possible to analyze all three apical views in all included patients? Were the results checked for intra- and inter-observer variability? Who were the examiners (one or several, level of expertise?) who recorded the images?

You are absolutely right that the image quality in essential for the measurement of myocardial strain and work. We included first 30 adolescents with anorexia nervosa, 30 obese adolescents and 34 controls. Due to their lower image quality or the lake of an apical view, we decided to exclude 7 patients. The recording of the patients included in the present study were of good quality, sufficient for the analysis of LV strains and myocardial work. In general, the image and cine-loops quality recorded on adolescent is high.

In accordance, we changed the manuscript in the method and results section:

“This prospective study included female adolescents with anorexia nervosa (AN patients, n=30), normal weight (n=34) and obesity (OB patients, n=30) aged 10–18 years”

And in the result section:

“Due to their lower image quality or the lake of an apical view, we decided to exclude 4 AN, 1 normal weight and 2 OB patients”.

To improve the reliability of data, all LV strains measurement were averaged on 3 to 5 cardiac cycles, the brachial artery measurements on 2 assessments obtained after a resting period of about 30 minutes. The myocardial work was assessed on each subject from an average of two analysis on two different cardiac cycles for each subject.

The intra-observer variability of LV strains analysis was previously conducted in our laboratory and was very good. In the present study, we evaluated the intra-observer variability of myocardial work, which appeared to be good (CV: 4.4%). This was added in the manuscript (statistical analysis paragraph):

“In the present study, the intraobserver variability for GMW was assessed on duplicate measurements on 60 subjects. The variability was very low, with a coefficient of variation of 4.4%.”

Concerning your last question, all resting echocardiography were recorded by only one very experienced echocardiographist, Stéphane Nottin, who practiced resting and effort echocardiographies for 20 years. Moreover, Stephane Nottin used 2D-strain echocardiography in his laboratory on all research protocols since 2006. All echocardiographic recordings were analysed a posteriori using EchoPac software by Justine Paysal, in relation with Stephane Nottin.

Line 128: There is a large different in duration of illness between the anorexia and obesity group (19 vs 130 months). As the authors write later, myocardial work reflects myocardial remodeling. If remodeling is one of the main targets of this analysis, the examined two diseases should have been in place for the same time, otherwise duration of illness can be suspected to be a significant confounder. Please comment on this.

You are absolutely right that the large difference in duration of illness between anorexia and obesity groups could be a significant confounder. This is why in the first version of our manuscript we added this limitation at the end of the discussion:

“Changes in myocardial function were mainly observed in obese patients, a result potentially explained by their higher duration of illness (and also of exposure to altered loading conditions) compared to AN patient.”

In the revised version, we added the following sentence:

“Further studies will be needed to assess regional LV strains and myocardial work in AN adult patient with a longer duration of illness”

Minors:

Lines 86f: please specify which ultrasound systems were used.

Done. According to your comment, we changed the following sentence:

“with Vivid ultrasound systems (GE Healthcare, Horten, Norway) using a 3.5 MHz transducer (M4S probe)”

In

“with Vivid Q ultrasound systems (GE Healthcare, Horten, Norway) using a 3.5 MHz transducer (M4S probe)”

Line 33: the well-established „impact“ on the cardiovascular system should be further addressed here: morbidity/mortality, long term effects.

According to your comment, we added the following sentences at the beginning of the introduction to further address the long-term effect of anorexia nervosa and obesity:

“During AN, cardiac features are frequent, reaching 80% in some studies. These complications range from morphological cardiac abnormalities to electrical abnormalities with a potential risk of sudden death (Fayssoil en 2021). In large epidemiological studies, obesity is associated with increased incidence of heart failure. Analysis of the Framingham Heart Study revealed that obese individuals had a doubling of the risk of heart failure over a mean follow-up of 14 years (Kenchaiah et al., 2002).”

What was the clinical state/ condition of the patients at time of inclusion? Outpatient clinic, inpatients? Were any of the anorexia patients on artificial nutrition? Refeeding syndrome?

All OB patients were recruited in a specific institution for obese adolescents, at the beginning of their one-year medical management and ongoing monitoring by a medical staff (diet and exercise training programs). AN patients were recruited at the hospital, either at the beginning of their medical care (n=16) or during their first months of refeeding period (n=10). However, their blood chemistry and liver function tests analysis indicated that none of them had a refeeding syndrome.

These specific points were added in the method section:

“All OB patients were recruited in a specific establishment for obese adolescents, at the beginning of their one-year medical management and ongoing monitoring by a medical staff. AN patients were recruited at the hospital, either at the beginning of their medical care (n=16) or during the first months of refeeding period (n=10). None of the AN adolescent reported purge and/or binge behaviour or had refeeding syndrome, as evidenced by their normal ionogram.”

Were there any blood results to be reported, NTproBNP?

Blood samples were obtained in our adolescent and according to your comment we added NTproBNP data in the methods and results sections:

In the methods section:

“A fasting venous blood sample was performed to biochemical determinations, with especially NT-proBNP assayed by automated immunoassay.”

In the results section:

“  NT-proBNP was higher in AN (79 ± 60 ng.L-1) compared to OB (34 ± 23 ng.L-1) and normal weight (39 ± 21 ng.L-1) patients (P<0.01)."

What types of anorexia were included, purge/binge etc.?

None of the AN adolescent reported purge and/or binge behaviour. This was also added in the method section:

“None of the AN adolescent reported purge and/or binge behaviour.”

Why did the authors choose to analyze a female only cohort, please comment on this.

We choose to analyse female cohort only since the prevalence of AN in male adolescent is very low (prevalence varies between 0.16 and 0.3% in men, compared to about 3% in females). This was added in the limitation section of the manuscript:

We choose to analyse female cohort only since the prevalence of AN in male adolescent is very low (prevalence varies between 0.16 and 0.3% in men, compared to about 3% in females)

Lines 127/128 „ as expected BMI was different between our groups”. This sentence is redundant as patients with anorexia and obesity were included.

According to your comment, we deleted this sentence.

Abstract, line 25, „reflecting specific regional myocardial remodeling“: this is only one possible explanation. Considering the small number of included patients the message should be phrased differently, e.g. “might reflect specific regional remodeling”.

The sentence has been changed according to your demand.

Reviewer 3 Report

The authors have studied LV mechanics in a very specific population, in view of the low number of publications on this subject, particularly on anorexia nervosa.

For obesity patient, authors might insist on the opposite results they have with literature. They quote Binnetoglü et al., but there were also Kulkarni A et al. Pediatric Obesity. 2018;13(9):541‑9. who share the same results : a GLS lower on obesity adolescents, where the population share quite same anthropometric data.

Haley JE et al. Pediatric Diabetes. 2020;21(2):243‑50. also report a lower GLS on obesity patients, but their population were older (23yo) and had only 32% of Caucasian, so these results were less comparable.

Then, there are also Sanchez AA et al. The Journal of Pediatrics. 1 mars 2015;166(3):660‑5. who described a lower GLS on their obesity population and some alteration on LV remodelling on 23% on their obese population, probably when duration of obesity was higher. They also observed intra- and inter-observer reproducibility.

Authors did not specify how many physicians were collecting the echocardiographic data. Is there a good inter-operators reproducibility? Was it assess, as well as intra-operator variability?

More recently, global LV myocardial work efficiency (GLVMWE), which combines the measurement of constructive and wasted work, provides information on global LV performance derived from the described and used pressure-strain curves. Could the authors try to compute this? See e.g. https://doi.org/10.1161/CIRCIMAGING.120.012072

But finally, we remain puzzled by this non-invasive estimation of LV pressure used. We understand the limitation that the authors stress of using brachial systolic pressure, and not the ascending aorta pressure. But we do not understand that they use reference 11 to say that it is overall working well…if I may quote that paper: “An imprecise pressure prediction…The mean value for all patient of RMSE was found equal to 12.3 mmHg for the estimated pressure wave curve. Differences were observed between maximum systolic values of measured and estimated pressures. This observation points to a flaw of this method: the arterial systolic pressure measure by a brachial cuff is imprecise and this imprecision grows when the arterial pressure is high. On top of it, arterial pressure could be false if the patient has brachial vascular disease.” Now, this is for the systolic LV pressure, but how do the authors explain how they think that their method can correctly assess the diastolic LV pressure? How can they consider that it will be the same for AN patients, or others with obesity?

Minor review :

  • Figure 2 : graph should be in better quality
  • Figure 3 : graph should be greater or in better quality.
  • L 192: “AN patients may have conserved” French false friend – you mean preserved I guess
  •  

Author Response

The authors have studied LV mechanics in a very specific population, in view of the low number of publications on this subject, particularly on anorexia nervosa.

 For obesity patient, authors might insist on the opposite results they have with literature. They quote Binnetoglü et al., but there were also Kulkarni A et al. Pediatric Obesity. 2018;13(9):541‑9. who share the same results : a GLS lower on obesity adolescents, where the population share quite same anthropometric data.

Haley JE et al. Pediatric Diabetes. 2020;21(2):243‑50. also report a lower GLS on obesity patients, but their population were older (23yo) and had only 32% of Caucasian, so these results were less comparable.

Then, there are also Sanchez AA et al. The Journal of Pediatrics. 1 mars 2015;166(3):660‑5. who described a lower GLS on their obesity population and some alteration on LV remodelling on 23% on their obese population, probably when duration of obesity was higher. They also observed intra- and inter-observer reproducibility.

Thank you for these references. We could observe that all these references reported lower values of GLS in obese populations. We added these references in our manuscript. We added these new references at the end of the following sentence:

“They have a normal GLS, a result inconsistently found since other studies described a decrease in GLS in children with obesity [17], [31], [32] [33]–[35].”

Authors did not specify how many physicians were collecting the echocardiographic data. Is there a good inter-operators reproducibility? Was it assess, as well as intra-operator variability?

We included first 30 adolescents with anorexia nervosa, 30 obese adolescents and 34 controls. Due to their lower image quality or the lake of an apical view, we decided to exclude 8 patients. The recording of the patients included in the present study were therefore all of good quality, sufficient for the analysis of LV strains and myocardial work. Moreover, to improve the reliability of data, all LV strains measurement were averaged on 3 to 5 cardiac cycles, and the myocardial work was assessed on each subject from an average of two analysis on two different cardiac cycles for each subjects.

All resting echocardiographies were recorded by only one very experienced echocardiographist, Stéphane Nottin, who practiced resting and effort echocardiographies since 20 years. Moreover, Stephane Nottin used 2D-strain echocardiography in his laboratory on all research protocols since 2006. All echocardiographic recordings were analysed a posteriori using Echopac software by Justine Paysal, in relation with Stephane Nottin.

The intra-observer variability of LV strains analysis was previously conducted in our laboratory and was very good (Doucende et al., 2010). In the present study, we evaluated the intra-observer variability of Global Myocardial Work on 60 subjects, which appeared to be very good: 1946 +/- 339 mmHg.% and 1927 +/- 336 mmHg.% for the first and the second evaluations, respectively (P=0.9) with a coefficient of variation of 4.4%.

We added the following sentences in the manuscript (statistical analysis paragraph):

“Intra-observer and inter-observer variability for 2D-strain analysis has been previously assessed in our laboratory, yielding maximal coefficient of variation values less than 8% for strains (Doucende et al., 2010). In the present study, the intraobserver variability for GMW was assessed on duplicate measurements on 60 subjects. The variability was very low, with a coefficient of variation of 4.4%.”

More recently, global LV myocardial work efficiency (GLVMWE), which combines the measurement of constructive and wasted work, provides information on global LV performance derived from the described and used pressure-strain curves. Could the authors try to compute this? See e.g. https://doi.org/10.1161/CIRCIMAGING.120.012072

Done. We added the data in the table 2. Moreover, we added the following sentences in the method section:

“Myocardial work efficiency (MWE) was calculated as previously described. (Samset)”

But finally, we remain puzzled by this non-invasive estimation of LV pressure used. We understand the limitation that the authors stress of using brachial systolic pressure, and not the ascending aorta pressure. But we do not understand that they use reference 11 to say that it is overall working well…if I may quote that paper: “An imprecise pressure prediction…The mean value for all patient of RMSE was found equal to 12.3 mmHg for the estimated pressure wave curve. Differences were observed between maximum systolic values of measured and estimated pressures. This observation points to a flaw of this method: the arterial systolic pressure measure by a brachial cuff is imprecise and this imprecision grows when the arterial pressure is high. On top of it, arterial pressure could be false if the patient has brachial vascular disease.” Now, this is for the systolic LV pressure, but how do the authors explain how they think that their method can correctly assess the diastolic LV pressure? How can they consider that it will be the same for AN patients, or others with obesity?

For evident ethical reasons, cardiac catheterization is absolutely unthinkable in obeses, anorexic or healthy adolescents, and thus the assessment of the LV intraventricular pressures during the cardiac cycle needs non-invasive methods.

You are right that Hubert et al. recently reported an imprecise pressure prediction with the method proposed by Russell et al.(2015). However, they also reported that the imprecision of the estimated curve is not equal along the cardiac cycle. In fact, the precision of the estimation of the pressure between AVO and AVC is quite excellent, with a RMSE of only 6.5 mmHg. This specific part of the curve is the period when the pressure is maximal and also the period when the LV strain is maximal. As myocardial work value resulted in pressure and strain, his value provides from this interval (i.e. AVC–AVO), when the estimation of the pressure is the best. This results in a precise work estimation.

Moreover, the validity of the method proposed by Russell et al.(2015) was confirmed in the experimental part of their study by an excellent correlation and a good agreement with loop area by invasive pressure – segment length analysis. Moreover, in the clinical part of their study, LV pressure – strain loop area using the non-invasive LV pressure curve showed a strong correlation and a good agreement with loop area using invasive LV pressure.

Minor review :

  • Figure 2 : graph should be in better quality
  • Figure 3 : graph should be greater or in better quality.

We checked carefully the quality of these figures in terms of dpi.

  • L 192: “AN patients may have conserved” French false friend – you mean preserved I guess

You are right, we changed "conserved" by "preserved"

Round 2

Reviewer 1 Report

The authors gave valid explanations to the observations made during the review and the work can to be accepted after minor corrections

Author Response

Dear Reviewer,

Our manuscript was corrected by an experienced external english native speaker (Richard Ryan, see in the revised manuscript).

Best regards

Reviewer 3 Report

It is very diplomatic to the reviewer to say "Similar values of GWE were also observed between groups", but it would be more interesting to give the values in table 2

Author Response

It is very diplomatic to the reviewer to say "Similar values of GWE were also observed between groups", but it would be more interesting to give the values in table 2.

Thank you for your comment. The correction was done in the last revised manuscript, but I make this correction in an additional file with table 2, and I forgot to add the new data in the manuscript with the following of the corrections. The GWE is now included in this file. (PAYSAL - Manuscript - R3)